# Low-Carb and Ketogenic Diets in Type 1 and Type 2 Diabetes

**DOI:** 10.3390/nu11050962

**Published:** 2019-04-26

**Authors:** Andrea Mario Bolla, Amelia Caretto, Andrea Laurenzi, Marina Scavini, Lorenzo Piemonti

**Affiliations:** Diabetes Research Institute, IRCCS San Raffaele Scientific Institute, Milan 20132, Italy; bolla.andreamario@hsr.it (A.M.B.); caretto.amelia@hsr.it (A.C.); laurenzi.andrea@hsr.it (A.L.); scavini.marina@hsr.it (M.S.)

**Keywords:** carbohydrates, ketogenic, diabetes, dietary patterns, nutritional intervention

## Abstract

Low-carb and ketogenic diets are popular among clinicians and patients, but the appropriateness of reducing carbohydrates intake in obese patients and in patients with diabetes is still debated. Studies in the literature are indeed controversial, possibly because these diets are generally poorly defined; this, together with the intrinsic complexity of dietary interventions, makes it difficult to compare results from different studies. Despite the evidence that reducing carbohydrates intake lowers body weight and, in patients with type 2 diabetes, improves glucose control, few data are available about sustainability, safety and efficacy in the long-term. In this review we explored the possible role of low-carb and ketogenic diets in the pathogenesis and management of type 2 diabetes and obesity. Furthermore, we also reviewed evidence of carbohydrates restriction in both pathogenesis of type 1 diabetes, through gut microbiota modification, and treatment of type 1 diabetes, addressing the legitimate concerns about the use of such diets in patients who are ketosis-prone and often have not completed their growth.

## 1. Introduction

A healthy diet is important for a healthy life, as stated by the old saying “You are what you eat”. This is even more important in today’s world where diabetes and obesity are pandemic. According to the International Diabetes Federation 8th Diabetes Atlas, about 425 million people worldwide have diabetes and, if the current trends continue, 629 million of people aged 20–79 will have diabetes by 2045 [1]. Nutrition is key for preventing type 2 diabetes (T2D) and obesity, but there are no evidence-based data defining the best dietary approach to prevent and treat these conditions.

In the last decades, low carbohydrate diets (LCD) and ketogenic diets (KD) have become widely known and popular ways to lose weight, not only within the scientific community, but also among the general public, with best-selling dedicated books or intense discussion on social media networks staying at the top of the diet trend list for years. These dietary approaches are effective for losing weight, but there is growing evidence suggesting that caution is needed, especially when these diets are followed for long periods of time, or by individuals of a very young age or with certain diseases [2,3].

In the past, when no insulin was available, LCD has been advocated as a treatment for type 1 diabetes (T1D), but the dietary recommendations of those times were quite different from the low carb/high fat diets recommended today [4]. Various diets with a low content of carbohydrates (CHO) have been proposed, such as the Atkins diet, the Zone diet, the South Beach diet and the Paleo diet [5]. The term LCD includes very heterogeneous nutritional regimens [6]; no univocal definition(s) have been proposed and clinical studies on LCD do often not provide information on CHO content and quality. For these reasons it is difficult to compare results from different scientific studies. The average diet CHO usually represents 45%–50% of daily macronutrient requirements, with “low carbohydrate” diets being those providing less than 45% of daily macronutrients in CHO [5]. According to some studies, LCD generally contain less than 100 g of CHO per day, with the overall macronutrient distribution being 50%–60% from fat, less than 30% from CHO, and 20%–30% from protein [7]. Very low carbohydrate diets (VLCD) are ketogenic diets with an even lower amount of carbohydrates, i.e., less than 50 g of carbohydrate per day [5], usually from non-starchy vegetables [8]. After few days of a drastically reduced consumption of carbohydrates the production of energy relies on burning fat, with an increased production of ketone bodies (KBs), i.e., acetoacetate, beta-hydroxybutyric acid and acetone; KBs represent a source of energy alternative to glucose for the central nervous system [9]. The increased production of ketones results in higher-than-normal circulating levels and this is why KD may be indicated for the treatment of refractory epilepsy [10,11], including children with glucose transporter 1 (GLUT1) deficiency [12]. People on ketogenic diets experience weight loss, because of lower insulin levels, a diuretic effect, and a decreased sense of hunger [6]. The most common negative acute effect is the “keto-flu”, a temporary condition with symptoms like lightheadness, dizziness, fatigue and constipation [6,8].

In view of the heterogeneity of available data, the aim of this review is to explore the possible role of low-carb and ketogenic diets in the pathogenesis and management of type 1 and type 2 diabetes.

## 2. Low-Carb and Ketogenic Diets in the Pathogenesis of Obesity and Type 2 Diabetes 

For decades, the pathogenesis of obesity has been explained as calories introduced in amounts exceeding energy expenditure [13]. More recently, the scientific discussion on the pathogenesis of obesity has focused on the question: “Is a calorie a calorie?”; in other words, whether the consumption of different types of food predisposes to weight gain independently of the number of calories consumed. According to a recent Endocrine Society statement [13], the answer to that question is “yes”, i.e., when calorie intake is held constant, body weight is not affected by changes in the amount and type of nutrients in the diet. However, it is known that the type of food impacts on the number of calories consumed, for example diets high in simple sugars and processed carbohydrates are usually high in calories and low in satiety-promoting fiber and other nutrients, favoring an increase in overall energy intake [13]. 

Some researchers [14] point out that the conventional model of obesity does not explain the obesity and metabolic diseases epidemic of the modern era. In a study by Leibel et al. [15], maintenance of a reduced or elevated body weight was associated with compensatory changes in energy expenditure and hunger, with the former declining while the latter has been increasing. These compensatory changes may account for the poor long-term efficacy of treatments for obesity, and understanding this physiological adaptation is of practical importance in order to approach the current obesity epidemic. 

According to an alternative view, dietary components have a main role in producing hormonal responses that cause obesity, and certain types of carbohydrate can alter the homeostatic mechanism that limits weight loss [14]. The carbohydrate-insulin model (CIM) of obesity hypothesizes that a high-carbohydrate/low-fat diet causes postprandial hyperinsulinemia that promotes fat deposition and decreases circulating metabolic fuels (glucose and lipids), thereby increasing hunger and slowing the whole-body metabolic rate. In this view, overeating is a consequence of increasing adiposity, rather than the primary cause. Insulin is the most potent anabolic hormone that promotes glucose uptake into tissues, suppresses release of fatty acid from adipose tissue, inhibits production of ketones from liver and stimulates fat and glycogen deposition. Dietary carbohydrates are the main driving force for insulin secretion and are heterogeneous in their glycemic index (GI) (an index of how fast blood glucose rises after their ingestion) [16], and glycemic load (GL) (derived from carbohydrate amount and glycemic index). The latter is the best predictor of post prandial blood glucose levels after CHO ingestion [17]. As carbohydrates are the main source of glucose, reducing their intake may lead to a decrease in insulin requirements, an improvement in insulin sensitivity and a reduction of post-prandial glycaemia [18]. In these terms, LCD may have a positive effect in the management of metabolic diseases and in the pathogenesis of obesity. 

In animal models, studies about the impact of LCD on metabolism and diabetes have yielded different and sometimes controversial results. In a mouse model, adult mice were fed isocaloric amounts of a control diet, LCD or KD, to determine the influence of different types of diet on longevity and healthspan [19]. The results showed that lifespan was increased in mice consuming a KD compared to those on a standard control diet, without a negative impact on aging [19]. In the study of Yamazaki and collaborators [20], in obese mice fed with very low-carb diet or isoenergetic low-fat diet (LFD), the authors found that both diets led to similar weight loss, but VLCD-fed mice showed increased serum concentration of fibroblast growth factor 21 (FGF21), ketone bodies, markers of browning of white adipose tissue, and activation in brown adipose tissue and hepatic lipogenesis. According to various studies on normal and diabetic rats, high GI diet promotes hyperinsulinemia, increased adiposity, lower energy expenditure and increased hunger [21,22,23,24]. In the study by Pawlak et al. [24], partially pancreatectomized rats were fed with high GI or low GI diets in a controlled manner to maintain the same mean body weight. Over time the high-GI group had greater increase in blood glucose and plasma insulin after oral glucose, lower plasma adiponectin concentrations, higher plasma triglyceride concentrations, severe disruption of islet-cell architecture and higher percent of body fat. By contrast, some data support a different hypothesis. In the study by Ellenbroek et al. [25], a long-term KD resulted in a reduced glucose tolerance that was associated with insufficient insulin secretion by β-cells. After 22 weeks, mice following a KD showed a reduced insulin-stimulated glucose uptake, and a reduction inβ-cell mass with an increased number of smaller islets, accompanied by a proinflammatory state with signs of hepatic steatosis.

Results of genetic studies are also controversial. In a recent report [26], bidirectional Mendelian randomization was used to test association between insulin secretion and body mass index (BMI) in humans. Higher genetically determined insulinemia was strongly associated with higher BMI, while higher genetically determined BMI was not associated with insulinemia. Moreover, in obese children it has been found that, in the early phase of obesity, alleles of the insulin gene variable number of tandem repeat (VNTR) locus are associated with different effects of body fatness on insulin secretion [27]. However, according to other studies in humans, even if genetic variants associated with body fat distribution are often involved in insulin signaling and adipocyte biology [28], genetic variants associated with total adiposity are principally related to central nervous system function [29]. Therefore, insulin-signaling pathways seem to have an impact on obesity pathogenesis, although they are not the only cause, allowing the rationale for other nutritional approaches different from LCD.

The hypothesis that carbohydrate-stimulated insulin secretion is the primary cause of common obesity, and metabolic diseases like T2D, via direct effects on adipocytes, seems difficult to reconcile with current evidence from observational and intervention studies [30]. In the DIRECT Trial [31], 322 obese subject (36 with diabetes) were randomly assigned to a low-fat/restricted-calorie, a Mediterranean/restricted-calorie or a low-carbohydrate/non–restricted-calorie diet. LCD was efficacious in reducing body weight, although it also caused a deterioration of the lipid profile, while the Mediterranean diet had a better effect on glucose control in individuals with diabetes. Similar results were reported by a recent meta-analysis [32], according to which persons on LCD experienced a greater reduction in body weight, but an increase in HDL and LDL cholesterol. In the larger Diogenes trial [33], a reduction in the GI of dietary carbohydrates helped maintenance of weight loss. Finally, the recent DIETFITS Trial [34] compared a healthy LFD with a healthy LCD and found no difference in weight change and no predictive value of baseline glucose-stimulated insulin secretion on weight loss response in obese subjects. In contrast with these data, Ebbeling et al. [35] reported that in 164 adults that were overweight or obese, total energy expenditure was significantly greater in participants randomly assigned to an LCD compared with high carbohydrate diet of similar protein content; pre-weight loss insulin secretion seemed to modulate the individual response to these diets. 

In summary, an increased CHO intake is important in the pathogenesis of obesity and T2D, although the role of additional factors still needs to be elucidated. 

## 3. Low-Carb and Ketogenic Diets in the General Population and for the Treatment of Obesity and Type 2 Diabetes

When considering the impact of LCD/KD in non-diabetic subjects, it is not possible to identify a univocal answer. The Prospective Urban Rural Epidemiology (PURE) study is a large, epidemiological cohort study, including more than 100,000 individuals, aged 35–70 years, in 18 countries [36]. Participants were followed for a median of 7.4 years, with the aim to assess the association between fats (total, saturated fatty acids, and unsaturated fats) and carbohydrate intake with overall mortality and cardiovascular events. The results showed that high carbohydrate intake (more than about 60% of daily energy) was associated with higher overall mortality and non-cardiovascular mortality, while higher fat intake was associated with lower overall mortality, non-cardiovascular mortality and stroke. Some experimental evidence from animal models provides a possible explanation for these findings, hypothesizing that the glucose-induced hyperinsulinemia, other than having negative metabolic effects, may also play a role in promoting malignant growth [37].

The PURE study findings were in contrast with the usual recommendation to limit total fat intake to less than 30% of total energy and saturated fat intake to less than 10%, and the authors even concluded suggesting a revision of dietary guidelines in light of their findings, promoting low-carb or ketogenic diets. However, it is important to remember that the PURE study is an observational study, and should not be interpreted as prove of causality [38]; secondly, the PURE study only provides information on the amount of total CHO intake, but not on the quality and source, and healthier macronutrients consumption was associated with decreased mortality [39,40]; and thirdly, the main sources of carbohydrates in low- and middle-income countries are mostly refined, indicating that the observed refined CHO consumption is likely a proxy for poverty [41].

On the other side, as described, in the DIETFITS randomized trial, no difference was observed in weight change between a healthy LFD and a healthy LCD (aiming to achieve maximal differentiation in intake of fats and carbohydrates, while maintaining equal treatment intensity and an emphasis on high-quality foods and beverages) in overweight/obese adults without diabetes after 12 months. As previous observations suggested a role of fasting glucose and fasting insulin as predictors for weight loss and weight loss maintenance when following diets with different composition in macronutrients [42], the DIETFITS study also tested whether a genotype pattern or insulin secretion were associated with the dietary effects on weight loss, but none of the two was. 

There is upcoming evidence that a higher focus should be placed on the quality and sources of carbohydrates as determinants of major health outcomes, rather than quantity [43]. A recent metanalysis described a U-shaped association between the proportion of CHO in diet and mortality: diets with both high and low percentage of CHO were associated with increased mortality, with the minimal risk observed at 50–55% of CHO intake [44]. Low carbohydrate dietary patterns favoring plant-derived protein and fat intake, from sources such as vegetables, nuts, peanut butter, and whole-grain breads, were associated with lower mortality, suggesting that the source of food notably modifies the association between CHO intake and mortality. Moreover, a recent series of systematic reviews and meta-analyses, supported by the World Health Organization (WHO), aimed to investigate the relationship between CHO quality (not total intake) and mortality and incidence of a wide range of non-communicable diseases and risk factors. Highest dietary fiber consumers, when compared to the lowest consumers, had a 15%–30% decrease in all-cause and cardiovascular mortality, and incidence of coronary heart disease, type 2 diabetes, and colorectal cancer and incidence and mortality from stroke; a significantly lower bodyweight, systolic blood pressure, and total cholesterol were also observed in high dietary fiber consumers [45].

Many studies support the positive effect of a low-carb diet in people with T2D. The study by Wang et al., compared the safety and efficacy of an LCD vs. an LFD in 56 patients with T2D in a Chinese population [46]; patients following an LCD achieved a greater reduction in HbA1c than those following an LFD, with no safety concerns. In another study, 115 obese adults with T2D were randomly assigned to a very-low-carbohydrate, high–unsaturated fat, low–saturated fat diet or to an isocaloric high-carbohydrate, low-fat diet for 52 weeks; both diets resulted in a decrease in body weight and an improvement in HbA1c, although without significant differences between the two groups. Moreover, the LCD achieved greater improvements in lipid profile (possibly explained by fat quality in the low-carb diet, which was high in unsaturated fat and low in saturated fat), blood glucose variability, and reduction of diabetes medication [47]. The same authors reported the longer-term (2-year) sustainability of these effects: after 2 years from randomization, there were no differences in treatment discontinuation between the 2 groups, and the results confirmed comparable weight loss and HbA1c reduction, with no adverse renal effects [48]. Interestingly, a low-glycemic/high-protein, but not a low-fat/high-carbohydrate diet was also proven to improve diastolic dysfunction in overweight T2D patients [49].

A further reduction in dietary carbohydrates, leading to ketosis, can be even more effective in T2D management. One non-randomized study compared the effects of a low-carb KD vs a “standard” low-calorie diet in 363 overweight and obese patients, of whom 102 had a diagnosis of T2D. A ketogenic diet was superior in improving metabolic control, even with a reduction in antidiabetic therapy [4]. In the study by Goday et al., 89 obese patients with T2D were randomized to a very low-calorie-ketogenic (<50 g daily CHO) diet or to a standard low-calorie diet for 4 months. The weight loss program based on a ketogenic diet was more effective in reducing body weight and in improving glycemic control, with safety and good tolerance [50]. A very-low calorie KD was also proven effective in 20 children (mean age 14.5 ± 0.4 years) with T2D following the diet for a mean of 60 days [51]. Since adherence to diet is important and requires frequent contacts with the patient (to verify the compliance and optimize antidiabetic therapy), some studies assessed, after a screening evaluation in the clinic, the feasibility, safety and efficacy of an online intervention. Saslow et al., after proving efficacy of a ketogenic diet in overweight and obese subjects with T2D or prediabetes with an in-person intervention [52], evaluated the efficacy of an on-line program and observed similar results to the in-person intervention [53]. Another group of investigators conducted an open-label, non-randomized, controlled study of a continuous care intervention (CCI, continuous remote care with medication management based on biometric feedback combined with the metabolic approach of nutritional ketosis for T2D management) compared to usual care. After 1 year, patients in the CCI group showed a better weight and glycemic control, reduced diabetes medication, significantly improved surrogates of NAFLD and advanced fibrosis, and improved biomarkers of cardiovascular (CV) risk, although observing an increase in LDL-cholesterol levels [54,55,56]; the CCI also documented long-term beneficial effects on some markers of diabetes and cardiometabolic health after 2 years [57].

One concern involves the relative lack of data about long-term safety, adherence and efficacy of LCD and KD in patients with diabetes [58]. We know that, for example, a Mediterranean diet is safe, can be maintained for a life-time and has durable effects on glycemic control when compared to a standard diet [59,60], in addition to reducing post-prandial lipemia [61]. Moreover, dietary approaches other than LCD and KD have been proven effective in T2D management. The Dietary Approaches to Stop Hypertension (DASH) diet was originally developed to prevent or treat high blood pressure, but had beneficial effects on glycemic control and cardiometabolic parameters of patients with T2D [62]. In the Look-AHEAD study an intensive lifestyle intervention, consisting of increased physical activity and reduced total and saturated fat intake, improved metabolic control and sometimes led to complete diabetes remission. According to some evidence, even a “high-carb” diet may be recommended in patients with T2D, if the diet is rich in fiber and has a low GI/GL ratio [63]. 

For all these reasons, the latest recommendations [3,64,65] do not indicate a unique eating pattern for people with diabetes, suggesting that meal planning and macronutrient distribution should be based on an individualized assessment of current eating patterns, preferences, and metabolic goals. A variety of dietary approaches is acceptable for the management of T2D and prediabetes, with emphasis placed on the importance of carbohydrate source; patients are suggested to prefer nutrient-dense carbohydrate sources that are high in fiber, to avoid sugar-sweetened beverages and to minimize the consumption of foods with added sugar.

Reducing carbohydrates intake is a helpful option but requires a regular periodic reassessment. Because LCD or KD results in ketosis, these meal plans are not suitable for some patients with T2D, including women who are pregnant or lactating, people with or at risk for eating disorders, or people with renal disease. Moreover, due to the increased risk of diabetic ketoacidosis (DKA), patients taking SGLT-2 inhibitors should avoid very-low-carbohydrate/ketogenic diets. [3,66].

In summary, the CHO source, in addition to the CHO amount, may have relevant effects on major health outcomes in the general population. With adequate patient selection and long-term monitoring, the reduction of CHO intake is effective in improving metabolic control in patients with T2D, with KD achieving stronger effects than LCD. Well-designed long-term studies on this topic are needed. 

## 4. Could Reducing Carbohydrate Intake Play a Role in the Pathogenesis of Type 1 Diabetes?

A normal gut homeostasis is the consequence of a fine-tuned balance between intestinal microbiota, gut permeability and mucosal immunity [67]. In this complex interplay, the alteration of one or more of these factors may contribute to the development and progression of inflammation or autoimmunity, that may result in diseases such as T1D or multiple sclerosis [68]. Gut microbiota plays a key role in gut homeostasis, and for this reason it is currently being so intensively investigated. Clostridia are mainly butyrate-producing and mucin-degrading bacteria, with immunomodulating properties, and are generally associated with a normal gut homeostasis [69,70,71]. De Goffau et al. [72] observed that β-cell autoimmunity is associated with a reduction in lactate-producing and butyrate-producing species, with an increased abundance of the Bacteroides genus. This finding agrees with what reported by Endesfelder et al., who suggested a protective role of butyrate in the development of anti-islet autoimmunity and onset of T1D [73]; furthermore, a reduced number of Clostridia was also observed in long-standing T1D patients [71].

It is known that diet influences gut microbiome [74] and that an acute change in diet alters microbial composition within just 24 h, with reversion to baseline within 48 h of diet discontinuation [75]. So how could a reduction in dietary carbohydrates, with a relative increase in fat or protein intake, affect gut microbiota and type 1 diabetes risk?

A “Western” dietary pattern, characterized by high fat and high salt intake, can induce alterations in gut microbiome, that affect IgA responses and lead to the production of autoantibodies. [68]. However, some studies have described that a high-fat diet is associated with a reduction in Bacteroidetes and an increased proportion of Firmicutes, both in mice and in humans [68,76,77,78,79], suggesting a potential protective role against the development of autoimmunity.

Conversely, some authors have described a reduced amount of short-chain fatty acids in subjects who consumed a diet high in animal protein, sugar, starch, and fat and low in fiber [80]. In another study, a high protein/low-carb diet was described to reduce Roseburia and Eubacterium rectale in gut microbiota, and lower butyrate in feces [81], thus resulting in a potentially unfavorable gut environment.

Another aspect to consider is whether different modes of dietary restrictions can play a role in the pathogenesis of T1D. Some studies indicate that both type and levels of nutrients can influence the generation, survival and function of lymphocytes and therefore can affect certain autoimmune diseases to some extent [82]. A fasting-mimicking diet (FMD) is a low-calorie, low-protein and low-carbohydrate, but high-fat 4-day diet that causes changes in the levels of specific growth factors, glucose, and ketone bodies similar to those caused by water-only fasting [83].

In mouse models cycles of FMD have been shown to promote the reprogramming of pancreatic islet cells, inducing a gene expression profile with similarities to that observed during fetal development. FMD cycles were also able to reverse insulin deficiency in mouse models of T1D and T2D [83], and to reverse insulin deficiency defects in human cells derived from T1D patients, indicating a potential ground for future studies [82,83]. 

In summary, gut microbiota likely has an important role in modulating the autoimmune process, possibly favoring autoimmunity in the presence of genetic predisposition and changes in diet. However, it is still unclear whether an LCD/KD may be protective against the development of anti-islet autoimmunity and prevent or delay the onset of T1D.

## 5. Low-Carb and Ketogenic Diets in the Treatment of Type 1 Diabetes

Prior to insulin discovery, strict low-carbohydrate diets with severe carbohydrate restriction (≤10 g/day) were the only available option to treat T1D [84]. Despite the many therapeutic advances achieved since those days, the management of T1D remains suboptimal in term of glycemic control [85]. Approaches that promote diet and insulin flexibility, such as Dose Adjustment For Normal Eating (DAFNE), are nowadays recommended by healthcare professionals [86]. Attention to food intake is required to calibrate at best the insulin dose prior to meals, so diet is an important tool in managing diabetes. As carbohydrates are the main responsible nutrient for post-prandial hyperglycemia [87], some authors reported benefits with carbohydrate restriction in patients with T1D, in term of both blood glucose fluctuations [88] and HbA1c levels [89]. There are several trials and some case reports about the use of LCD in T1D; unfortunately, these studies are very heterogeneous and it is difficult to compare their results [90]. In children with medically refractory epilepsy and T1D, the use of KD can be a hazard due to the risk of severe ketoacidosis, but some reports in literature suggested that this diet was safe and efficacious in reducing seizures in the long-term [91,92,93].

In the small randomized trial by Krebs et al. [89] ten adult patients with T1D were randomized to a standard diet (without restrictions, mean patients CHO intake was 203 ± 92 g/day) with carbohydrate counting or to a restricted carbohydrate diet (75 g of carbohydrates/day) plus carbohydrate counting. After 12 weeks, the group on LCD had significant reductions in HbA1c and daily insulin doses and a non-significant reduction in body weight and no changes in glycemic variability. In contrast, in an observational study on 11 adult patients with T1D who followed a KD (<55 g of carbohydrates), the KD was associated with good HbA1c levels and reduced glucose variability, but also with dyslipidemia and an increased frequency of hypoglycemic events [94]. In the case report by Toth C. et al. [95], ketogenic paleolithic diet was proposed in a 19-years-old male with newly diagnosed T1DM and resulted in normalization of glucose levels, increased C-peptide levels and increased triglycerides and LDL cholesterol. It is worth to note that in this case report there is no mention about ketone bodies level range; moreover, C-peptide level increase was documented only 2 months after diagnosis, when it is not so uncommon to observe a rise in C-peptide levels (honeymoon phase) [96].

An issue about the use of LCD can be the long-term tolerability. In many cases LCD was stopped before 1–2 years for a variety of reasons, often because intolerable and with a limited choice of foods [91,92,93]. In the case series by De Bock and colleagues [97], carbohydrate restriction in growing children led to anthropometrical deficits, higher cardiovascular risk metabolic profile and fatigue. A clinical audit performed to assess the long-term adherence to LCD in people with T1D showed that after two years about half of the people ceased adhering, the other half adhered for at least four years, with the adherent patients experiencing a sustained reduction in HbA1c levels [98].

Lennerz et al., performed a survey [99] recruiting volunteers from an online community for people with T1D who follow a very-low-carbohydrate diet, as recommended in the book by Dr Bernstein’s for diabetes management [100]. Patients self-reported a very good outcome in term of HbA1c (5.67% ± 0.66%) and a low incidence of side effects, with a mean duration of 2.2 ± 3.9 years on this LCD diet. There is an obvious selection bias because this community is not representative of the general T1D population, but represents a self-selected group of people who voluntarily follow this nutritional approach. Moreover, only 17% of the web community participants responded to the survey and only 8% of them provided medical data. We could speculate that patients following low-carb diets have a high attention to meal composition, the same level probably encountered in patients who apply precise carbohydrate counting. It would be reasonable to design a randomized clinical trial comparing patients on LCD with patients precisely applying carbohydrate counting, rather than with general population of patients with T1D.

Long-term outcomes of KD in patients with T1D, especially children and adolescents, are unknown [101]. Moreover, there is no consensus on the acceptable level of ketosis in patients with T1D when on a KD [101]. Nowadays we have the possibility to measure blood ketones using dedicated meters or urinary ketones using reactive stripes. A new tool, the breath acetone sensor, will hopefully allow the easy monitoring of LCD safety [102]. 

In the latest Standards of Medical Care by American Diabetes Association, KD and LCD, although very popular among patients, are not included in the medical nutrition therapy recommendations for T1D [3]. 

In summary, LCD may be an option for short-term improvement of glycemic variability in some patients with T1D, although we recognize the limited evidence-based knowledge in this field, which truly needs well-designed trials about the long-term safety and efficacy of LCD. 

## 6. Conclusions

Reducing CHO intake with an LCD is effective in reducing body weight and, in patients with type 2 diabetes, improving glycemic control, with a stronger effect with a very low carb diet (KD). However, LCD and KD may not be appropriate for all individuals. Especially in patients with type 2 diabetes, it is necessary to balance the potential increase in cardiovascular risk because of the unfavorable lipid profile observed with KD with the benefits deriving from weight loss and improvement of glycemic control. Moreover, long-term compliance with low-CHO diets is still an issue. 

In type 1 diabetes, there is no present evidence that an LCD or a KD can delay or prevent the onset of the disease. These diets have the potential to improve metabolic control, but caution is needed because of the risk of DKA, of worsening the lipid profile and, in children, the unknown impact on growth. 

Even in studies in the general population where a higher CHO intake was associated with worse outcomes, healthier macronutrients consumption was associated with decreased cardiovascular and non-cardiovascular mortality. When healthy LFD was compared to healthy LCD, good results in terms of weight loss were observed with both diets. Therefore, macronutrients source, i.e., CHO quality, are not negligible factors, and preferring fibers and nutrient-rich foods is a good option for everyone. For this reason, when designing future studies on nutrition, it will be important to evaluate not only the amount of CHO, but also their type. 

Even though this review is not about exercise, we want to underline in the conclusion that diet and exercise are both vitally important to good health in diabetes. All the exercise in the world will not help you lose weight if your nutrition levels are out of control, but the adoption and maintenance of physical activity are critical foci for blood glucose management and overall health in individuals with diabetes and prediabetes. In this direction, we reported the conclusion of the recent position statement of the American Diabetes Association [103]: “Physical activity and exercise should be recommended and prescribed to all individuals with diabetes as part of management of glycemic control and overall health. Specific recommendations and precautions will vary by the type of diabetes, age, activity done, and presence of diabetes-related health complications. Recommendations should be tailored to meet the specific needs of each individual…”.

In conclusion, LCD and KD can be effective options in patients with obesity and/or type 2 diabetes, although they are not the only available dietary approach for such patients. In any diet, LCD and KD should be tailored to individual needs and patients should be followed for an extended period of time. The use of those diets in patients with type 1 diabetes is still controversial and their long-term safety is still unproven.

Further large-scale, long-term, well-designed randomized trials are needed on this topic to assess the long-term safety, efficacy and compliance of reducing dietary CHO in patients with diabetes, and particularly with type 1 diabetes of all ages, and to find the best dietary composition as for glycemic control, weight loss, and CV risk in all patients with diabetes.

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
