# Peer review of "Low-Carb and Ketogenic Diets in Type 1 and Type 2 Diabetes"

_nutrients, 2019, doi:10.3390/nu11050962_

Reviewer 1 Report

This is a comprehensive and easy to read review on a very popular topic. I have only very few suggestions/comments:

line 70: it is unclear, what the "yes" relates to. This could be misinterpreted as the sentence directly preceding the "yes" has the opposite meaning.

it would be helpful to have a short summarizing paragraph at the end of each of the 5 sections.

Author Response

Milan, April 18th, 2019

Dr. Matthew Cooke

Guest Editor for the Special Issue “Precision Nutrition and Metabolic Disease”

Nutrients

Cc: Ms. Aria Chen

Assistant Editor

Nutrients

Dear Dr. Cooke,

We thank the Editor and the Reviewers for their careful reading and fruitful comments to our manuscript.

We thank the Reviewer for the interest in our paper and for the pertinent comments raised. We addressed her/his remarks as follows.

“Line 70: it is unclear, what the “yes” relates to. This could be misinterpreted as the sentence directly preceding the “yes” has the opposite meaning.”

We thank the Reviewer for reporting this possible misinterpretation, and have amended the text as follows (page 2 line 71): According to a recent Endocrine Society statement [13], the answer to that question is “yes”, i.e, when calorie intake is held constant, body weight is not affected by changes in the amount and type of nutrients in the diet.

 “It would be helpful to have a short summarizing paragraph at the end of each of the 5 sections”

We take the point of the Reviewer and consequently better summarized each section with a short paragraph.

Page 2 line 63. In view of the heterogeneity of available data, the aim of this review is to explore the possible role of low-carb and ketogenic diets in the pathogenesis and management of type 1 and type 2 diabetes.

Page 4 line 148. In summary, an increased CHO intake is important in the pathogenesis of obesity and T2D, although the role of additional factors still needs to be elucidated.

Page 6 line 255. In summary, CHO source, in addition to CHO amount, may have relevant effects on major health outcomes in the general population. With adequate patient selection and long-term monitoring, the reduction of CHO intake is effective in improving metabolic control in patients with T2D, with KD achieving stronger effects than LCD. Well-designed long-term studies on this topic are needed.

Page 6 line 298. In summary, gut microbiota likely has an important role in modulating the autoimmune process, possibly favoring autoimmunity in the presence of genetic predisposition and changes in diet. However, it is still unclear whether a LCD/KD may be protective against the development of anti-islet autoimmunity and prevent or delay the onset of T1D.

Page 8 line 354. In the latest Standards of Medical Care by American Diabetes Association, KD and LCD, although very popular among patients, are not included in the medical nutrition therapy recommendations for T1D [3]. In summary, LCD may be an option for short-term improvement of glycemic variability in some patients with T1D, although we recognize the limited evidence-based knowledge in this field, which truly needs well-designed trials about the long-term safety and efficacy of LCD.

In the revised manuscript the text changes are highlighted in red.

Thank you for your support and consideration.

Sincerely,

Lorenzo Piemonti, MD

Professor of Endocrinology

Diabetes Research Institute

IRCCS Ospedale San Raffaele

Via Olgettina, 60 – 20132 Milan, Italy

Phone: 0039 02 2643 2706

Fax: 0039 02 2643 2871

piemonti.lorenzo@hsr.it

Reviewer 2 Report

I think this is a very well written manuscript . The information is applicable to the population at large since more than 20% of the population is obese in the US. I am not 100% sure if this is the exact percentage but I know it is high.  I am not an advocate of high

protein diets and I think that this manuscript discusses some long range health concerns associated with ketogenic diets. I am not an advocate of low carb, high fat diets either and these diets are also discussed. Diets such as the Atkins diet, the South Beach diet etc. can be extremely dangerous in the long run.  I also like the support of "Is a calorie a calorie?" by the Endocrine Society.  Your paper was not about exercise but since I am an exercise physiologist, I would have liked to see exercise as a recommendation or a future topic for review in the conclusion.  Exercise changes your metabolism and is the only way that fat weight can be lost in conjunction with a balanced diet.   I would like you to add something about exercise and weight loss in a section called future directions or something of this nature.  It could not be in the conclusion since you did not discuss exercise in the paper

Author Response

Milan, April 18th, 2019

Dr. Matthew Cooke

Guest Editor for the Special Issue “Precision Nutrition and Metabolic Disease”

Nutrients

Cc: Ms. Aria Chen

Assistant Editor

Nutrients

Dear Dr. Cooke,

We thank the Editor and the Reviewers for their careful reading and fruitful comments to our manuscript.

We thank the Reviewer for the interest in our paper and for the pertinent comments raised. We addressed her/his remarks as follows.

As the reviewer also underlined, our paper was not about the exercise. This does not mean that excercise is not relevant. For sure exercise should require a dedicated review and this could be a future topic for other review. We recognize the vale of the reviewer recall and in the conclusion section of revised paper there is a short but relevant claim on the value of exercise.

In the revised manuscript the text changes are highlighted in red.

Thank you for your support and consideration.

Sincerely,

Lorenzo Piemonti, MD

Professor of Endocrinology

Diabetes Research Institute

IRCCS Ospedale San Raffaele

Via Olgettina, 60 – 20132 Milan, Italy

Phone: 0039 02 2643 2706

Fax: 0039 02 2643 2871

piemonti.lorenzo@hsr.it